

# Dietary effects on fitness in captive-reared Hawaiian tree snails

Evan Strouse[1], Melissa R. Price[1] and David R. Sischo[2]

[1] Department of Natural Resources and Environmental Management, University of Hawai'i, Honolulu, Hawai'i, United States
[2] Department of Land and Natural Resources, Division of Forestry and Wildlife, Honolulu, Hawai'i, United States

Corresponding author
Evan Strouse,
evanstrouse7@gmail.com

## ABSTRACT

The native terrestrial snail fauna of the Hawaiian Islands faces numerous threats that have led to severe range reductions, population declines, and extinction of species. With the continued declines of many wild populations, a crucial component of preserving Hawaiian terrestrial snail biodiversity is through captive rearing programs, like that implemented by the Hawai'i Department of Land and Natural Resources Snail Extinction Prevention Program. Rare and endangered tree snails in the family Achatinellidae, which feed on epiphytic microbial communities, are maintained in captivity with a diet that includes native vegetation brought in from nearby forests, as well as a cultured fungus originally isolated from native host trees. Recent mortality events in lab populations have been attributed to wild-gathered vegetation. These events have increased interest in developing a completely manufactured or cultured diet that would eliminate the need for exposure to wild-gathered plants. This study compared survival and egg production in *Auriculella diaphana* provided with lab-cultured fungus, and those provided with wild vegetation. We compared the number of eggs laid and number of deaths among three treatments: (1) wild collected vegetation only; (2) wild vegetation supplemented with laboratory-cultured fungus; and (3) laboratory cultured fungus only. Mortality did not significantly differ among treatments, but the number of eggs laid was significantly higher in snails provided wild vegetation and cultured fungus ($F = 24.998$; $P < 0.001$), compared with those provided with only wild vegetation ($t = 1.88$, $P = 0.032$) or only cultured fungus ($t = 4.530$, $P = 0.004$). Our results suggest: (1) the existing strain of cultured fungus alone is not sufficient to maintain captive-reared snail populations; (2) the additional energy or calcium provided by the cultured fungus appears to enhance egg reproduction in captive-reared populations; (3) the presence or absence of live vegetation influences snail behavior, including aestivation and egg laying. These results highlight the importance of ongoing research to culture additional species of fungi at a rate that could support captive-reared populations, as the diversity of fungi present in wild epiphytic microbial communities may be important for snail reproductive health.

## INTRODUCTION

Under threat from a variety of disturbances, such as climate change and introduced species, imperiled animals and plants are increasingly brought to ex situ facilities for captive propagation as a hedge against extinction (*Snyder et al.,1996*). Despite immediate protection from the in situ drivers of extinction, ex situ captive propagation of endangered species faces many challenges including, but not limited to, the maintenance of genetic diversity, overcoming genetic and behavioral adaptation to captivity, the creation of appropriate diets, as well as exposure to and transmission of diseases and parasites (*Ralls & Meadows, 2001*; *Price & Hadfield, 2014*; *Price et al., 2015*). For many species, information on natural history is lacking, and thus basic husbandry practices must be tested and developed either on dwindling populations of endangered species, or on proxy species with more stable populations.

The Hawaiian Islands once hosted more than 750 species of terrestrial snails (*Cowie, Evenhuis & Christensen, 1995*). This remarkable diversity occurring over such a small land area arguably makes these islands one of the greatest land-snail hot-spots on earth. Hawaiian terrestrial snails or "kāhuli", once ubiquitous across multiple islands, were the inspiration for Hawaiian chants and storytelling, denoting romance and good omens, and were associated with the cool mists of the mountain forests (*Sato, Price & Vaughan, 2018*). Despite their abundance and relevance to local culture and lore, during the last century threats including shell collection, habitat loss, climate change and the introduction of invasive predators, have decimated this fauna (*Hadfield, 1986*; *Solem, 1990*; *Hadfield, Miller & Carwile, 1993*; *Hadfield & Saufler, 2009*; *Holland, Montgomery & Costello, 2010*, *Yeung & Hayes, 2018*; *Gerlach et al., 2020*). Most of the species persisting are now only found in small portions of their historical ranges, with many species requiring ex situ captive propagation to prevent extinction.

One subfamily, Achatinellinae, has been the focus of conservation efforts, including captive rearing, for over 30 years (*Hadfield, Holland & Olival, 2004*). Snails in this subfamily have long lifespans, late maturity, and produce a single offspring per birth, making them highly susceptible to extinction (*Hadfield, 1986*; *Hadfield & Miller, 1989*; *Hadfield, Miller & Carwile, 1993*). Due to extreme declines the entire genus *Achatinella*, endemic to Oʻahu, as well as two species of *Partulina* endemic to Lanaʻi, and one species of *Newcombia* endemic to Maui, have been listed as Endangered under the U.S. Endangered Species Act (*U.S. Fish and Wildlife Service, 1981*; *U.S. Fish and Wildlife Service, 2013*), while many other species across the 13 families present in the islands remain unlisted but extremely imperiled.

To support conservation efforts, the Snail Extinction Prevention Program (SEPP), was established by the Hawaiʻi Department of Land and Natural Resources. One strategy of the program is to use captive propagation to safeguard species from extinction, maintain evolutionary potential, and generate individuals for reintroduction to protected areas in the wild (*Price et al., 2021*), Currently, the SEPP captive rearing facility maintains populations of 38 species in eight genera, from five islands. The captive snails are housed in

population-specific terrariums inside environmental chambers that control relative humidity, temperature, light, and simulated rain.

In the wild, tree snails feed on diverse epiphytic fungi that grow on native plants (*Kobayashi & Hadfield, 1996*; *O'Rorke et al., 2015*; *Price et al., 2016*). Captive-reared tree snails are provided fresh leafy branches of native plants from forested areas on O'ahu. In addition, the snail diet is supplemented with a lab-cultured fungus in the genus *Cladosporium*. This is a naturally occurring biofilm component isolated from wild leaves and detected in fecal samples of wild snails (*O'Rorke et al., 2015*). The fungi are inoculated on potato-dextrose agar and infused with calcium carbonate powder to support shell growth. Prior research has demonstrated that some snail species grow at a faster rate when supplemented with lab-cultured *Cladosporium* sp. (*Kobayashi & Hadfield, 1996*).

Recent mortality events above baseline levels in laboratory populations of Hawaiian tree snails were attributed to a possible pathogen introduction, or exposure to toxicants or environmental toxins (*Price et al., 2015*; *Sischo et al., 2016*). One of the main routes for pathogen introduction, or toxicant/toxin exposure in the laboratory is through the continued collection and use of leaves from the wild to feed captive-reared snails. Fresh vegetation is provided to captive-reared snails bi-weekly. Snails in the laboratory are maintained in high densities, and contaminated vegetation has the potential to introduce disease or parasites that could quickly spread within, and potentially among, terrariums. Furthermore, diets containing native vegetation preclude zoos and other institutions outside of Hawai'i from participating in Hawaiian land snail conservation. This is problematic as Hawai'i is prone to natural disasters such as hurricanes and tsunamis. The disbursement of critically imperiled snail species amongst multiple institutions outside of the islands eliminates the proverbial all eggs in one basket scenario.

The International Partula Program rears rare snail species in the genus *Partula* from the South Pacific region in participating zoological societies and research institutions across Europe and the United States (*Pearce-Kelly et al., 1997*). Their strategy of spreading vulnerable populations among many institutions has proven successful at preventing extinctions and serves as a model for our efforts here in Hawai'i. However, one of the keys to their success has been the development of a manufactured diet, allowing snails from islands in the South Pacific to be reared anywhere in the world (*Clarke, 2019*). Similarly, *Gilbertson, Rundell & Niver (2019)* describes captive rearing success with the endangered ovate amber snail *Novisuccinea chittenangoensis*, after optimizing a captive diet that includes dried leaves of preferred plants. Moving forward, it is critical for our conservation efforts here in Hawai'i to develop and test dietary options that increase reproduction and survival and are not dependent upon fresh native vegetation.

As a foundational step in this effort, in this study we compared the mortality and egg production of captive tree snails provided with a diet of cultured fungus, with those fed on native vegetation, to elucidate how the diet of the snails affects survival and egg production. If snails provided with a cultured or manufactured diet could maintain fitness equivalent to snails fed on wild-gathered vegetation, then Hawaiian tree snails could be reared anywhere in the world, with a reduced potential for pathogen introduction or

exposure to manmade toxicants such as herbicide etc., and toxins such as secondary metabolites of naturally occurring bacteria and fungi.

## MATERIALS & METHODS

We observed the number of eggs laid and mortality in adult snails collected from wild populations and captive-reared over a period of 13 weeks in the captive rearing facility run by the Snail Extinction Prevention Program. The focal species selected for the study, *Auriculella diaphana*, is a native tree snail in the subfamily Auriculellinae, a sister subfamily to that of Achatinellinae. The subfamily Achatinellinae contains all of the Hawaiian land snails listed as Endangered under the US Endangered Species Act of 1973 (*Holland & Hadfield, 2004*), but low numbers in all remaining species, as well as low fecundity and long maturity times, do not allow for efficient evaluation of the impact of diet on fitness. *Auriculella diaphana* was selected as a proxy for these closely related endangered species, due to an epiphytic diet similar to endangered snails in the Achatinellinae subfamily (*Pilsbry & Cooke, 1912*; *O'Rorke et al., 2014*; *Price et al., 2017*). Further, oviparity in *A. diaphana* allowed for timely comparison of fecundity among treatments. However, as eggs of *A. diaphana* take over 1 month to hatch, overall reproductive success was beyond the scope of this study; egg production was used to evaluate the potential impact of the diet treatment on reproductive success.

### Sample Collection

A total of 120 *A. diaphana* were collected from the Ko'olau mountains on the island of O'ahu, Hawai'i, USA under the supervision of the Department of Land and Natural Resources–Snail Extinction Prevention Program (SEPP). The snails were hand-collected from multiple trees and shrubs across the range of the study population. Because some achatinellid species have very small ranges, in some instances spending their whole lives in a single tree (*Hadfield & Saufler, 2009*), distance was used as a proxy for genetic diversity. Host trees at this particular site were all non-native species to Hawai'i, including *Cestrum nocturnum*, *Bischofia javanica*, *Coffea arabica, and Hedychium coronarium*. All snails selected were adults (sexually mature). Adults were identified by a lip formation on the shell aperture indicative of sexual maturity (*Pilsbry & Cooke, 1912*). Once collected, the snails were immediately transported to the captive-rearing facility in small plastic containers.

### Acclimation Period

At the laboratory, snails were randomly placed in four large clear plastic terrariums, at a density of 30 individuals per terrarium, for a 3-week period. This allowed the snails to acclimate to their new environment, prior to initiating the study. During acclimation all snails were offered both the wild vegetation and lab-cultured fungus. While acclimating, all eggs deposited in the terrariums were removed during cleaning and were not included in the study.

## Setup

Terrariums were housed in a single Caron brand insect-growth environmental chamber (Model 6025-1). Day and nighttime temperature and relative humidity were controlled to mimic wild conditions (21 °C and 80% relative humidity during day simulation, and 20 °C and 90% relative humidity during night simulation). An automatic misting system simulated rain periodically through the lid of the terrarium. A 12-h day and 12-h night cycle was created, with florescent chamber lights, to simulate the average photoperiod in the Hawaiian Islands. Throughout the study, each terrarium was cleaned and censused once every 2 weeks. This process involved removing the old vegetation and/or cultured fungus and replacing it with fresh material, steam sterilizing the cage in a dishwasher, accounting for all adult snails, checking for deaths, and searching for eggs. To confirm that the presence or absence of vegetation within different treatments did not alter the ambient temperature and relative humidity within terraria, a temperature and humidity data logger was placed within one terrarium of each treatment, and no differences were detected.

## Experimental Design

A total of ninety individuals were selected from the acclimated snails and randomly assigned to three feeding treatments for 10 weeks: (1) wild collected vegetation; (2) wild vegetation and laboratory-cultured fungus; (3) laboratory cultured fungus only. Vegetation for all laboratory snails at the SEPP lab is collected weekly from several field sites in the Waianae and Koolau mountain ranges on Oʻahu. This fresh vegetation was provided to all treatments, except the fungus only treatment, and consisted of native species in the following percentages 80% *Metrosideros polymorpha* and a mixture of approximately 20% of *Freycinetia arborea*, *Nestegis sandwicensis*, *Pisonia umbellifera*, *Psydrax odorata*, and *Antidesma platyphyllum*. Each treatment consisted of three terrariums with 10 snails per terrarium. All of the terrariums were cleaned and checked for deaths and eggs laid as described above. The eggs and dead snails were removed from cages during each cleaning and recorded. Fecundity was measured as the number of eggs produced per adult per 2 weeks. Differences in the number of eggs laid among treatments was assessed with an analysis of variance (ANOVA) using JASP, version 0.9.2. If a difference among treatments was detected with an ANOVA, *t*-tests were run to identify which treatments differed from others. Additionally, a repeated measures analysis of variance test was performed using JASP to determine if egg production in each cage changed significantly over time.

## RESULTS

There were significant differences in egg production among all treatments ($F = 24.998$; $P < 0.001$) (Fig. 1). Snails in Treatment 1 (vegetation and fungus) produced 2.4 times more eggs (241 eggs laid, average of 8 eggs/snail) than snails in Treatment 2 (vegetation alone) (102 eggs laid, average of 3.4 eggs/snail) ($t = 1.88$, $P = 0.032$). Snails in Treatment 1 produced 30.1 times more eggs than Treatment 3 (fungus alone) (8 eggs laid, average of 0.27 eggs/snail) ($t = 4.530$, $P = 0.004$). Snails in Treatment 2 (vegetation alone) produced 12.8 times more eggs than Treatment 3 (fungus alone) ($t = 2.642$, $P = 0.046$). The snails

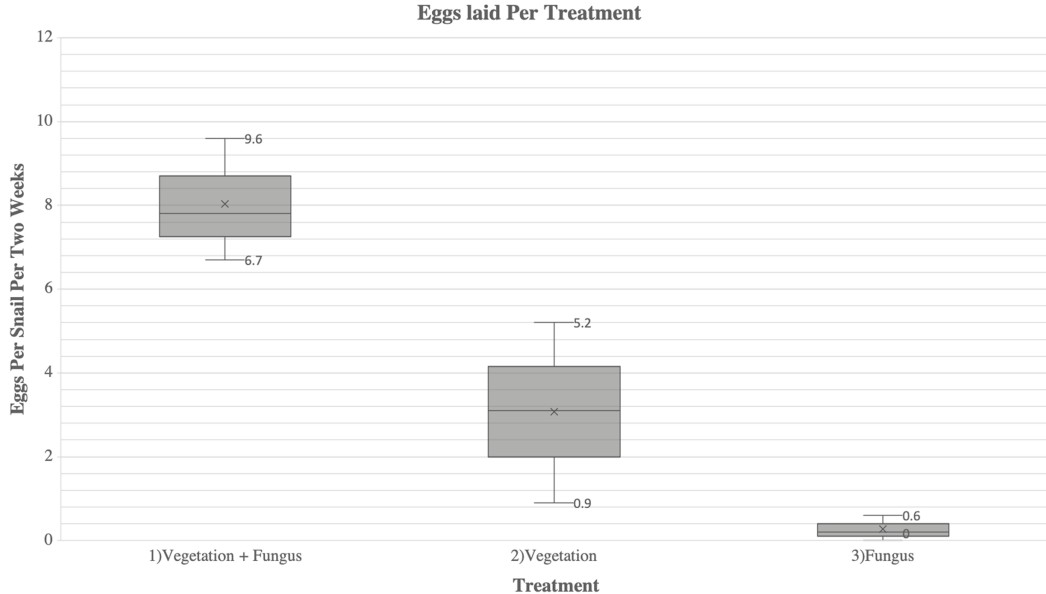

**Figure 1** **A comparison of fecundity between each treatment of *A. diaphana*.** Each box-plot shows the average eggs laid per snail per every 2 weeks (represented as "x"). The minimum and maximum eggs laid per snail per 2 weeks for each treatment are represented by the numbered lines.

in Treatment 3 (fungus alone) appeared to be less active throughout the study and were observed to be sealed to the sides or top of the cage when cages were cleaned, a behavior referred to as "aestivation". There was no significant difference in egg production over time ($F$ (8, 3) = 2.150, $P$ = 0.120).

Overall survivorship was high in all treatments with only four deaths total during the study. Of these deaths, three of them occurred in Treatment 1 (vegetation and fungus), and one occurred in Treatment 2 (vegetation). There was no mortality in Treatment 3 (fungus alone).

## DISCUSSION

Maintaining low mortality, as well as high fecundity, in captive-reared populations is particularly important for endangered species across taxa. Snail species in the subfamily Achatinellinae do not reach maturity until 3 to 5 years of age and produce fewer than ten offspring per year (*Hadfield & Miller, 1989*; *Hadfield & Mountain, 1980*; *Price et al., 2015*). Therefore, optimizing captive-rearing conditions is crucial for Hawaiian tree snail conservation, as small differences in survival and reproduction can have large impacts on the long-term viability of captive-reared colonies. The primary objective of this study was to compare survival and egg production between captive-reared snails fed a diet of a natural epiphytic biofilm on leaves collected from native forests, and captive-reared snails fed a diet of lab-cultured fungus. The number of eggs laid was significantly higher in snails provided a diet of both cultured fungus and wild-gathered leaves, than in snails provided a diet of leaves alone or fungus alone. Further, snails provided with only native vegetation laid more eggs than snails provided with lab-cultured fungus alone. These

results suggest that a diet consisting of diverse microbes, like that occurring on native vegetation, may be important for snail reproduction as both treatments including vegetation yielded more eggs than snails provided with fungus alone. Additionally, because snails offered vegetation and fungus laid more eggs than the other two treatments, it is possible that bulk (defined here as fungal mass) and or calcium supplementation may be important factors in snail reproduction in captivity. Snails also readily consume the potato dextrose agar, which may increase reproduction simply by providing more calories (*O'Rorke et al., 2015*). Another potential explanation is that surface cues on leaf or branch substrates may be necessary for egg production.

While the mortality in this experiment was minimal, the length of the study period may not have revealed long-term differences in survival among treatments. Each time the snails were censused, little to no activity was seen in the snails in Treatment 3 (fungus alone). In contrast, snails in the other two treatments were more active throughout the trial, moving around the cage. The aestivation behavior in the snails fed only the lab-cultured fungus may be a stress response that would result in higher mortality over a longer interval due to long-term sub-lethal stress. It is possible that snails in the fungus-only treatment were less active due to a lack of suitable substrate rather than an insufficiency in their diet. Relative humidity and temperature were uniform throughout the cages; however, leafy branches may hold more surface moisture than an empty plastic cage. It is possible that the presence or absence of plant material impacts snail behavior that in turn impacts reproduction. However, a previous study of *Auriculella diaphana* comparing differences in egg production between snails maintained on native vegetation verses non-native vegetation (*Holland, Chiaverano & Howard, 2017*) found that snails fed native vegetation had 20.5-fold higher number of eggs produced than snails maintained on non-native vegetation. Non-native plants may host less diversity or less mass of native fungal species but would likely provide suitable substrate for other behaviors. These studies highlight the critical need to evaluate nutrient and energy needs for captive-reared snails, and the potential for other epiphytic fungal species derived from native plants to support thriving captive populations.

The results of this study complement previous work suggesting the current diet for captive-reared snails, which includes wild vegetation in addition to lab-cultured fungus, is optimal to maximize reproduction and survival during captive-rearing compared to diets consisting of cultured fungus or leaves alone. Previous studies have identified thousands of bacterial and fungal operational taxonomic units that are on leaf surfaces and ingested by snails (*O'Rorke et al., 2015*; *Price et al., 2015*). Our study may suggest that the single *Cladosporium* sp. cultured in the laboratory is insufficient to support captive populations on its own, but its continued use as a supplement to wild-collected vegetation is supported as beneficial, as previous studies have suggested (*Kobayashi & Hadfield, 1996*). While this study was short in duration, it has been particularly valuable in directing future lines of inquiry. Follow-up research should include: (1) determining the nutrients associated with diverse epiphytic communities ingested by snails to support the creation of a manufactured diet; (2) identifying which of the many epiphytic microbes ingested by snails supports long-term survival and reproduction; (3) development of

methods to culture and grow additional fungal species at a rate that will support captive-rearing of snails; (4) identifying which microbes produce secondary metabolites that may be toxic to snails and associated with snail mortality, and if they vary in distribution, time, and density geographically so they could be avoided; (5). whether the presence of substrate (such as leafy branches in this study) impacts snail behavior that in turn impacts survival and reproduction.

## CONCLUSION

Hawaiian land snail conservation currently relies on captive propagation to prevent extinction in multiple species. The current fungus cultured for supplementing captive snail diets is not solely sufficient to support adequate reproduction and survival, but its continued use to supplement wild vegetation is recommended until a more diverse set of cultured fungi can be developed. Further research regarding dietary impacts on captive snails, and the development of additional varieties of cultured fungi, is timely and urgent since the risk of introducing a pathogen, parasite, toxin, or toxicant from vegetation collected from the wild has the potential to decimate captive populations, potentially resulting in extinctions. Further, there are few, if any studies of nutritional requirements to support the development of snail diets. If a completely manufactured or cultured diet that supports snail growth and reproduction can be identified, the risk of exposure could be minimized, and captive propagation of Hawaiian tree snails could then be conducted anywhere in the world.

## ACKNOWLEDGEMENTS

We also appreciate the valuable input from members of the Wildlife Ecology laboratory at the University of Hawaiʻi at Mānoa.

### Funding

This work was funded by the Undergraduate Research Opportunities Program at the University of Hawaiʻi at Mānoa. This research was supported by the Hawaiʻi Department of Land and Natural Resources-Division of Forestry and Wildlife, and the Snail Extinction Prevention Program. The funders had no role in study design, data collection and analysis, decision to publish, or preparation of the manuscript.

### Grant Disclosures

The following grant information was disclosed by the authors:
University of Hawaiʻi at Mānoa.
Hawaiʻi Department of Land and Natural Resources-Division of Forestry and Wildlife.

### Competing Interests

The authors declare that they have no competing interests.

## Author Contributions

- Evan Strouse conceived and designed the experiments, performed the experiments, analyzed the data, prepared figures and/or tables, and approved the final draft.
- Melissa R. Price analyzed the data, authored or reviewed drafts of the paper, and approved the final draft.
- David R. Sischo analyzed the data, authored or reviewed drafts of the paper, and approved the final draft.

## Data Availability

The data are available in the Supplemental File.

## Supplemental Information

Supplemental information for this article can be found online at http://dx.doi.org/10.7717/peerj.11789#supplemental-information.

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
