# Peer review of "Dietary effects on fitness in captive-reared Hawaiian tree snails"

_PeerJ, doi:10.7717/peerj.11789_

## Round 0.1 · original submission · Major Revisions

· Academic Editor

Major Revisions

Thank you for your submission.

It has been reviewed by two expert reviewers. Both have provided a long list of suggestions to improve the manuscript. Of particular note, both reviewers would like the authors to discuss the results of their feeding trial in more detail. Some additional supplemental data may be necessary.

Both reviewers agreed that the paper is an important contribution. With that in mind, please read through their comments carefully and address each one.

Reviewer 1 ·

Basic reporting

no comment

Experimental design

Some minor additions to Methods are suggested in the General Comments to Authors section.

Validity of the findings

Conclusions are well-stated, but I think there could be some further exploration regarding why the cultured fungus only treatment was so much less preferred to the leaves and fungus combo. I've elaborated a bit on this in the General Comments.

Additional comments

Summary:

The authors use a "less endangered" species of Hawaiian tree snail to try to optimize the captive diet of Hawaiian tree snails in their captive breeding program. This is of critical importance because so many achatinellid species have already gone extinct. The authors are under particular pressure here to find a diet that enhances snail survival and reproduction, as well as one that could potentially be replicated elsewhere in the world, since otherwise they are stuck continuing to breed these snails in one isolated locality that is prone to the potentially devastating effects of power outages due to hurricanes and storms, as well as disease or toxicants (which they specifically mention) that could come in on the wild vegetation they are currently using to feed their snails. Overall this is a straightforward study with straightforward results, that suggest that that magical food solution that would allow them to raise endangered tree snails outside of Hawaii, has not yet been found. The snails barely move or grow if raised on cultured fungus alone.

This research is important because of its subject matter (highly endangered snails) and these snails' current precipitous decline (to only a few individuals of certain species; here again I would mention this with specific examples-- you can't assume all of the readers will have heard about the species SEPP has or the numbers left. It adds to the weight and importance of the work). This study also would provide peer-reviewed evidence that these snails differ from the partulid tree snails, another group of highly endangered Pacific island snails that, in contrast, have been able to be raised in many different localities on diet that humans can concoct easily with a published recipe. (Along these lines, I think it would be helpful for the authors to mention these partulid breeding programs and a few citations, because it would help show the general interest in the study, as well as help contrast with the authors' results. This is the only similar Pacific island land snail breeding program of equivalent size out there.).

The paper is clearly written and interesting, particularly to researchers, conservation managers and zookeepers focused on increasing numbers of understudied, small-bodied endangered species through captive breeding programs.

Specific comments by line number:

Lines 48-49: Use of "radiation" too vague here: The 750-plus snails are the result of multiple radiations. But it is definitely true that the resulting diversity is among the greatest on Earth! (also I would suggest changing "world" to "Earth" in this radiation context, just because world tends to refer to human matters.

94-95: Suggesting elaborating slightly on this statement to say that this is problematic because it could affect the security of critically endangered species (e.g. if a hurricane wiped out the SEPP lab).

101: Remove "drastically." Unnecessary.

106: Redefine the "SEPP" abbreviation and where it is located. Or if you don't want to do it twice, maybe save this explanation for Methods section. (Also not sure why this was done at line 119 instead of earlier.)

108: By "sister subfamily" I am assuming you mean phylogenetically speaking? If so, a citation is called for, especially since later you are talking about the fact that you use Auriculella as a proxy _because_ it is closely related.

113: "auriculellines" should be lower case. You only need upper case for the full subfamily name.

120: What tree species?

121: Since people reading this might be more familiar with just about any other animal besides tree snails (i.e. animals with larger home ranges for individuals), it might be worth noting through a citation the fact that most of these tree snails likely do not leave their home trees. I don't know it that is yet known for Auriculella in particular, but still you can give some scientific basis for the reason you took the multi-tree approach, based on another achatinellid species.

137: Add something about this 12:12 photoperiod being chosen to mimic Hawaii day/night cycle.

146: What species comprise the "wild collected vegetation"? Are these all native species? Is this a secret for a reason or can you share at least some of this. For depressing practical reasons, if this species declines or goes extinct later, it is important to have a record of what species you were finding it on. And do the species tested differ from the species you typically find this snails species on? If so, why?

156, 158: I think the better way of writing "2.4x" would be "2.4 times" just because it could get confused with equation notation in other papers.

161: How sad!

177: I have a question about the idea of "bulk" in the snail diet suggested here. I am not sure that this conclusion necessarily follows from the Results. Or, you might just need to talk more about to what extent having leaves or branches of leaves (should also specify in Methods), increases the surface area for fungi/microbes. I am wondering about your result showing that snails raised on cultured fungus alone, just appear inactive and don't eat much. Could it be that the winning combo of leaves plus cultured fungus has as much to do with snails being stimulated to eat through the presence of their natural leafy habitat, than it does with the cultured fungus being a poor food? I am just wondering if what snails are getting out of the leaves plus cultured fungus diet is not just about quantity-- and perhaps suggesting that it is hard to say. It just seems strange that the snails in the cultured-fungus only tank didn't really seem active at all-- so it wasn't just that they were forced to be eating less because of lack of microbe quantity. For example, it is also possible that the leaves/branches provide so many more natural-feeling perching sites, that snails are bound to be more active, and perhaps cross over and through many more microbes just as a result of tasting familiar foods on their branches, and then happening by the cultured fungus when they are on their feeding route. I suppose that still could be a matter of survival and repro. "success due to bulk"-- but the reason for it could be less about the bulk itself, and instead some other x-factor that the presence of leaves/branches provides.

184: I think you are partially getting at my comment above here with the mention of the "stress response." But I think you need to go into more detail about what you mean. A vet could assess signs of stress in an animal, even a snail. Sometimes this involves retraction into the shell, etc.-- it depends on the stimulus causing the stress. So it might be more helpful to explain the typical activity level of these snails, and how the behavior you noticed differed from that, a bit more in the Results, and/or add any references about stress response or adaptation to e.g. dry periods in snails. Could the lack of leaves/branches in the terraria also have reduced humidity to the point where the snails would not want to be active? And thus they might not have eaten for that reason?

185-189: Suggest moving this idea to the top of your Discussion, since it helps underscore the general importance of the study. You aren't just testing a bunch of diets for fun, you are actually trying to save some species. Also I would suggest breaking this up into two separate sentences, so the 3-5 year thing is separated out. The link between the two ideas in the sentence isn't so closely tied together that it all needs to be in one sentence. This might come off as more convincing if you break it up.

195: "Our" study, rather than "this" study, so it is clear you aren't talking about one of the referenced studies.

199: Can you give some examples of pathogens, parasites or toxicants that you know are occurring and/or that are a problem now, or could be more so in the future? Also if forests become drier due to climate change, could such contaminants become more concentrated on the leaves?

215: Not necessarily here, but maybe elsewhere in the discussion, could you give some examples from snails or other organisms where such dispersed breeding efforts have proven more successful than breeding in just one location, and why? It would be good to explain the increased risk involved by having all individuals of a critically endangered species in one place. Also, presumably decreasing or mitigating such risk is also part of the USFWS species management plan (cite)?

·

Basic reporting

1. I would like the introduction to give a broader contextualisation of the literature concerning raising snail in ex situ facilities. I have provided a few links to some studies that aren’t cited, but there are more. In particular the Partula literature. I know that grazing strategies in Partula are somewhat diverse, and some of the results of the Tahitian snail studies are old (but successful), and the more current references (last 2 -3 years) are in the “grey” literature. However, there are instructive scientific publications that bolster some of the authors more speculative points (Gouveia et al. 2011 https://bit.ly/3kJQW0a, and https://bit.ly/3kDYl0M). Of the literature that works with A. diaphana, I think that Holland et al., 2017 (https://bit.ly/33VRNnB) should at least be cited, which uses the same response variable and same snail species in a similar experiment. However, I think the authors would make their publication stronger if they argue that they are extending on the findings of the Holland et al., publication.

2. A large portion of the Introduction and Discussion place the motivation for addressing diet as being due to the risk of introducing pathogens on branches/leaves that are periodically added to the terraria for the snails to graze epiphytic microbes from. I think that this might be too speculative for the Introduction, as a pathogen has not been identified. In L88 “Recent mortalities above baseline levels in laboratory populations of Hawaiian tree snails were attributed to a possible pathogen introduction or exposure to an environmental toxicant (Price et al. 2015; Sischo et al., 2016)” the cited papers only really speculate on the cause of population crashes, and are population genetic studies, not epidemiological. Price et al. 2015 refer to an unsuccessful search using microscopy to look for Microsporidia as a cause of mortality, but found none. I agree with the authors that pathogens are a likely explanation of population crashes amongst different species, but I think that other less likely causes are possible too and shouldn’t be ruled out without evidence.

3. I also think that there are many other routes of pathogen transmission other than wild leaves – the ex situ facilities aren’t high-level containment facilities. If they had UV treated hepa-filtered air, directional airflow away from the incubators, controlled human access and strict PPE, then perhaps the inference that the only path a pathogen could be introduced is through the leaf cuttings would be more convincing. There is strong evidence that Partula snails that are fed artificial diets also have crashes and problems with pathogens (Cunningham & Daszak 1998: https://bit.ly/360w0xF; also https://tinyurl.com/yxoypj5u), and to my knowledge, Koch’s postulates are almost impossible to satisfy with endangered taxa, and suspected pathogens are routinely found in healthy animals (Cunningham et al 1996, https://bit.ly/2RTwn4J). These points all make definitive attribution of pathogen mediated disease difficult. I agree with the authors that bringing cut foliage into the facility has a lot of problems. However, I don’t think that this particular inference, that they harbour pathogens, has sufficient evidence.
a. I wonder if the pathogen angle is needed. Perhaps it is sufficient motivation that the experiment addresses whether culture can be improved? I am fine with some level of speculation in the Discussion, but with some caveats.

Experimental design

1. Egg production vs egg viability. Currently the number of “eggs produced” is used as the independent variable in the ANOVAs. It would really improve this study if some measure of the condition of the eggs was used as a response variable instead. In particular, did the authors record how many eggs hatched? Or even a crude measure, such as egg size?

If hatching or egg condition has not been recorded, then this limitation should be acknowledged. Also, the authors should acknowldge that a 10 week study has limited scope and an ideal study would track survival and reproduction of offspring.

2. A problem which would be easy to remedy is that the authors use an ANOVA (L 152), but they should use a repeated measures ANOVA because they are resampling the same terrariums over time. That is, without using a repeated measure design they violate the statistical test’s assumption of independence and creates a ‘batch’ effect because the measure at each timepoint is not of an independent batch.

3. The choice of a box and whisker plot compounds this problem. It would be preferable to plot egg production as a function of time as a line for each terrarium (using 3 different colours, one for each treatment), that way we could see if production is constant, goes up or goes down, across the course of the experiment both within and between treatments.

4. The species of tree cuttings need to be identified, as well as when they were collected (i.e., were they fresh when put in?) and from where. This is for at least four reasons that impact the replicability of this study. 1. Wilting occurs at different rates for different plant taxa, just like a vase of different flowers, some plants will wilt in a day, and others might appear to hold their structure for the full two weeks before the plants are changed. This could change the behaviour of snails, it is also likely to change the microbial communities that depend on different aspects of the living plant host’s physiology and metabolism. 2. Rotting will likewise occur at different rates with different plant hosts, changing the relative proportions of microbes associated with decomposition and differentially liberating various nutrients and micronutrients from the different species of decaying plants. 3. Snails seem to prefer particular host tress and might behave differently if they are on a preferred host (Price et al. 2017). 4. Similarly, Holland et al., 2017 (https://bit.ly/33VRNnB), find the highest egg production for A diaphana occurs on native host plants (interestingly, both this study and Holland et al. 2017, report similar levels of egg production).

5. Humidity and temperature loggers were deployed in the experiment to check that these variables were identical across treatments, but the results are not presented. These could be relevant in interpreting if treatment effects were confounded. For example, Hadfield Holland and Olival 2004 (https://bit.ly/33QMxl7) claim that humidity measurements are lower in terraria that lack leaves (page 23), which would explain non-feeding behaviour in that treatment group. Did the authors add any substrate to the terrariums such as paper (following the Partula protocol https://tinyurl.com/yxoypj5u)?

6. I think it might be helpful for the authors and journal if they identify the protocols that were in place to ensure the snails in the fungus on PDA trial that stopped feeding were not impacted in the long run.

Validity of the findings

1. The (easily addressable) major criticism that I have of the manuscript is that it draws very specific, but underdetermined, inferences from the experiment. This is an experiment that appears simple, comparing the presence or absence of two things, and would therefore be expected to have a simple interpretation. However, they -- are very heterogenous things, which makes the interpretation a little messy, which I think comes through in the Abstract which talks about microbial diversity on the leaves (L30) or energy in the agar/fungus supplement (L32). Discussion, especially in the inference that “the likely need for both microbial diversity and bulk in snail diets, and the potential benefit of calcium supplementation (L176-L177)” as well as lines 191-195 concerning diversity.
I think that a lot of these points are plausible, but the experiment only tells us that a combination of native leaves and a fungus monoculture grown on a carbohydrate + calcium rich agar produce more snail eggs than on leaves alone. The resolution of the experiment has no finer scale of information or gradient regarding microbial diversity, energy, micronutrients or bulk.
I do like the agenda-setting sentences on lines 200-206, and maybe it is sufficient to list the follow up experiments that need to be performed.

Additional comments

First of all, I like the general approach of trying to understand the ecology of endangered snail species experimentally by using snails that have a life history strategy that makes them more resilient to population fluctuations. By using such proxy species, the research on imperilled snails is able to move away from speculation and actually experimentally test hypotheses. I do have some criticisms of this particular piece of research, but I feel that the research project as a whole is very important and I hope that my feedback can help the authors make the most of the work they have done.

Some jargon is treated as synonymous when, although terms have overlapping meaning, they do not mean identical things. I am thinking of the interchangeable use of the words: egg production, growth, reproduction, fecundity, fitness. In this study, I think egg production is the best term to use.
L26 give t and p values for significant associations
L33 Replace “growth and reproduction” with “egg production”
L38-and elsewhere: It is conventional to italicise latin phrases
L90 I think toxicant is more often used for an artificially introduced toxin, would “toxin” be a better word?
L106 I don’t think the snails were observed for 16 weeks. Below it says they were held for 3 weeks acclimation (L127) and experimentally observed for 10 weeks (L146)
L112 I think Price et al 2015 is supposed to say Price et al 2017
L112 neither of these two studies establish that A diaphana are abundant species in the wild. Some of that information is in the grey literature. And, neither Price et al nor O’Rorke et al establish that their diets overlap. The best reference for comparing the diets of snail species is still possibly Pilsbry, Hyatt and Cooke 1912. O’Rorke et al 2015 does look at the diet of two Auriculella ambusta individuals, but not A diaphana. If the authors want to be more quantitative O’Rorke et al 2017 (https://go.nature.com/3iXmP4P) do sequence microbes from the feces and phyllosphere of A diaphana and these, along with the sequences from the O’Rorke et al 2015 and Price et al 2017 study are publicly available, so the authors could be more thorough and compare the diversity of these. However, this isn’t necessary, just an option.
L120 What were the host trees - Cestrum nocturnum? Price et al 2017 suggests the identity of the host might be relevant.
L129 What species of plant cuttings were put in the terraria? Was it the same as the snails were collected from or different? If Different, what species? Was the foliage sourced from the same location for the duration of acclimation and experimental timeframe?
L148 Were all terrariums in the same incubator?
L175 “These data suggest that a diet consisting of diverse microbes, like that occurring on native vegetation, may be important for snail reproduction.” I think the authors should add that other factors might be responsible for this though – eg habitat rather than food cues.
L176 What about the possibility that the habitat induces/facilitates egg production?
L177 I’m not 100% sure what “bulk” means here. Is the inference that the “plant only” terrarium might be food limited?
L182 “. . . the length of the study period may not have revealed long-term differences in survival among treatments”. The authors currently only refer to the long term impacts of aestivation. Personally, I would expand on this point about expanding the trial to last for longer. For example, although the snails provided with fungus on PDA laid more eggs, we know that these snails come from an energy poor environment and therefore providing them with a starch+sugar rich substrate (PD agar) has unpredictable outcomes for their long-term health, resilience and survival. However, I don’t insist on this as I understand that the authors might disagree with my point of view.
L185 Or starvation?
L192 I’m not sure it can be said it is “optimal”, it just appears to be the best set up for egg productivity. There might be an experimental set up that is as yet untested that is optimal. Also, long term mortality and reproductive fitness might be better in another treatment.
L200-206 I like this agenda setting paragraph. Possibly the authors should add the need to test short/long term components of an artificial diet given L204
L205 I would prefer “secondary metabolites” or even “toxin” to “environmental toxicants”. The latter term is frequently used to refer to exogenous toxins, rather than microbial metabolic products.

---

## Round 0.2 · Minor Revisions

· Academic Editor

Minor Revisions

First, I apologize immensely for the delay between reviews. Second, I have read through your responses to the reviewers and it appears you've addressed all major issues they raised. I thank you for your hard work on this front. The review was looked at by an additional reviewer and they raised some minor points that I think could be quickly addressed in a revision.

Thank you again for your patience in waiting for this response. I look forward to a re-submission!

Reviewer 1 ·

Basic reporting

yes

Experimental design

yes

Validity of the findings

Yes, but also please see specific comments in the "general comments to author" section. Thank you.

Additional comments

96-98: There is not necessarily a cause-effect relationship here. Yes, when snails are at high densities the potential for disease spread would be greater. But the issue of a disease first being brought in from the wild is a separate issue, that would be a problem whether or not the snails are at high densities in the lab. I suggest separating out these different factors into different sentences.

106: Suggest removing "very." "Successful" is enough-- especially since with Platydemus and E. rosea encroaching in the wild, it is probably already a stretch to say that extinction prevention has been fully successful, unfortunately.

109-111: This miss-states what Gilbertson et al. (2019) say, and/or maybe the statement appears misleading because it follows a mention of Clarke's manufactured diet. Gilbertson's diet by contrast 1) isn't manufactured, 2) it isn't really a captive diet in the sense you are talking about in the paper, and 3) it definitely isn't perfected. Gilbertson et al. refer to wild leaves being brought in, with some other stuff to supplement. So really it is closer to the Hawaiian tree snail lab's original diet of leaves brought in from the wild periodically. Again: unfortunately. Gilbertson's study focuses mainly on which wild species are better, in terms of which leaf species the snails grow best on.

120: It is unclear what "pathogen" and "toxicants and toxins" refer to here. Are these compounds produced by the wild fungi and trees/plants naturally? Are they chemicals that you worry might have been sprayed onto the wild vegetation, or introduced through rainfall or air pollution? Are the pathogens tree diseases or snail diseases? Since Ohia lehua diseases have made the news recently, this might be especially confusing. After reading O'Rorke's reviewer 2 comments and your responses (e.g. response to #3) I was surprised that this part about toxins or pathogens was left for readers to wonder about. People are already chemical-averse, but it doesn't mean that any toxin you bring in isn't something the snails also already experience in the wild and might therefore already be adapted to. It would really help I think to give some specific examples that you think might be issues for the snails, and then also note that this is speculative. I just think that someone who hasn't worked in your lab and talked to the people you are talking to about these snails every day, might have no idea what the real or perceived concerns are with respect to "toxicants" and "pathogens." There are bacteria and viruses all around us, even when we try our best (as you have) to remove them. Perhaps the problem is with impaired immune response in captive snails, more so than the fact that contaminants are coming in. That might even make sense in a crowded terrarium with rotting vegetation and lack of wind exposure (not a criticism here-- that would be true of any captive breeding setting for snails!). Again, quite speculative here, but I think it is important to say so when you really don't know. That is OK by me. Honestly, even your speculation would be helpful, because it gives potential avenues for future study. And sometimes it is better to keep the discussion broad so that alternatives aren't accidentally missed (and that is with the caveat that yes, with 30 years of research in the rearview, I understand that you feel like the conversation has exhausted many possibilities already).

204: Replace "less than" with "fewer than"

211: The result that the addition of cultured fungus is interesting in light of Reviewer O'Rorke's comment regarding PD agar, in his reference to L182. While I do understand that achatinellids in the lab have been fed wild leaves/branches, fungus, and the agar on which the fungus grows for 30 years-- I do think it is still worth a sentence or two speculating in the discussion along the lines that O'Rorke has mentioned, and perhaps then saying exactly what you said-- that the PD agar has been an important part of the diet in captivity for a long time. Readers might not realize that snails eat the agar as well as the fungus growing on top. They might also not be aware that rainforests are nutrient-poor environments. Maybe there is no reason to suspect that snails directly consuming agar over decades in captivity is a problem-- but for readers 20 years from now reading your paper, and wanting to disentangle different effects of food types, as we become more advanced in captive rearing of invertebrates, I think this would be helpful. O'Rorke: " although the snails provided with fungus on PDA laid more eggs, we know that these snails come from an energy poor environment and therefore providing them with a starch+sugar rich substrate (PD agar) has unpredictable outcomes for their long-term health, resilience and survival."
I would also add that even with 30 years of experience, this is still but one example, out of fewer than five, ten? major endangered land snail captive breeding efforts in the world. So anything you can say, even if a hunch, or even if you disagree, can be very useful for the next study in another land snail group in another setting. I would urge the editor to be a bit lenient in allowing you to speculate and/or post disagreements to this reviewer's comment in the Discussion, which I think is preferable to eliminating possibilities without strong evidence. I don't think it would detract from your paper, it would just help to ensure that it provides a balanced view, given the results and the difficulty in interpreting them.

---

## Round 0.3 · Minor Revisions

· Academic Editor

Minor Revisions

Thank you for your revision and for your patience in awaiting a response.

I can see that the authors have addressed many of the minor issues raised by reviewers. However, there is still a pressing concern about some of the statistics used by the authors.

I thank you for your work in improving the clarity of the manuscript but feel that this issue raised by a previous reviewer needs to be addressed.

·

Basic reporting

The manuscript is well written

Experimental design

See below

Validity of the findings

See below

Additional comments

This is a topic where any new data is very useful. The authors have either made changes or given reasons why they won’t change the manuscript, which I am happy with and I think the manuscript is fine for publication except on two points. However on those two points, I would strongly recommend that the authors reconsider their approach given the reasons that I raised earlier, but I’ll more fully explain why this is below.

First, the choice of statistics for the experimental design. I still think the ANOVA and t-test are inappropriate tests because the data violates assumptions of those tests. Namely, the response variables are not independent of each other (and it would be worth viewing the distribution of the residuals). By using an ANOVA they are treating each level as though there are 15 independent results for each level, when there are only 3. Nested inside each of those 3 is a repeated measurement of the same snail cage. At the very least this needs to be treated as a blocked design. I think it is a repeated measures design. I understand that authors disagree with this because they aren’t interested in time, but it is still a component of the experimental design and needs to be accounted for. Considering their response, I think the best approach would be a mixed effects model, then the authors can perform an AIC to see if it is fair to remove time from the model. Alternatively they could (maybe) do a test for variation with time and a blocked design test sequentially. Previously I suggested a repeated measures ANOVA because conceptually it isn’t too different from what the authors have done. They just need to check the distribution of the residuals whichever approach they choose.

Some examples of why it is important to test if time should be included/excluded from the model: although egg production is generally higher in the plant-cutting + fungus_&_agar treatment this productivity goes down in at the end of the experiment in one terrarium. In contrast, in two out of the three terraria that contain cuttings the egg production is zero or less than ten for the initial four or more weeks and then counts shoot up to over ten. These look like a temporal pattern, and if it is, it needs to be accounted for. The third of these terraria dries out, so that compounds this problem.

For the graph, for the same reason, I don’t think the authors can use the box and whisker plot of all the data binned into three explanatory treatments. A line graph of egg production vs time with three colours (one for each treatment) with one line for each snail enclosure shows ALL the information of the experiment in just nine lines. The current graph obscures a lot. Also, I think the y-axis label needs to be checked.

Second, lines 91-93 “Recent mortality events above baseline levels in laboratory populations of Hawaiian tree snails were attributed to a possible pathogen introduction, or exposure to toxicants or environmental toxins (Price et al. 2015; Sischo et al., 2016, Sischo et al., in prep).”

I am happy with the rest of this paragraph, the gist of which is that plant cuttings from numerous wild trees increase the risk of pathogens entering the facility. I strongly agree that this is possible. However, I think that it is misleading to state that snails deaths have been “attributed” to a possible pathogen, toxin or toxicant and then cite two population genetics studies that do not test for pathogens or toxins. Price et al. (2015) includes a pers com that microsporidia tests had negative results, but that is all. I think this sentence should be removed. I worry that future generations of scientists will read it and maybe not explore a new hypothesis that could transform snail conservation because of the confidence with which this sentence is written and that it is backed by citations that don’t back it up at all.

The Sischo et al., in prep, result sound very useful, but as non-peer reviewed literature need to be moved to the discussion. Furthermore, if referred to, then the nature of this work needs to be elaborated on a little (given that we can’t read it elsewhere yet). I don’t think the authors need to give the results – it sounds like it was a lot of work and it is fair that it stand as its own manuscript. But the reader needs some vague indication of what the study involved, eg “histology of prematurely dead specimens showed lesions not in other specimens” (I think that is what the study is about, I’m guessing from the authors comments) and tell us if known pathogens were isolated that might cause morbidity and if Koch’s postulates were attempted or not and whether they were satisfied. Then cite (Sischo et al., unpublished)” or a BioRxiv preprint reference. It sounds like there is some really positive stuff to come.

Otherwise, I am happy with the manuscript, as I said, all information on this topic is welcome. The above comments are long, but that only reflects that this is a reply to my earlier feedback and I am trying to make my reasoning more clear. Some minor points below:

Line 221 “may increase reproduction” I’d say “have unknown effects”. Increased nutrient and energy has massively variable effects on invertebrate egg production and fitness.

Line 245 change “optimal” to “better”, there isn’t enough evidence yet to say what is optimal

L246 “and survival” I’d delete this, survival is outside the scope of this experiment

L251 “beneficial” to “beneficial for increased egg production”

---

## Round 0.4 · accepted · Accept

· Academic Editor

Accept

Thank you for addressing the comments raised by the final reviewer. I see that you have gone through their comments in detail and added additional analyses to address specific concerns. I very much appreciate your time and effort.